# Mechanistic study on the sulfate migration in glycosaminoglycans during MS fragmentation

Lukasz Polewski [1,2,5], Murat Yaman [3,4,5], Matko Tokić[1], Mateusz Marianski [3,4] ✉ & Kevin Pagel [1,2] ✉

Glycosaminoglycans use positional sulfation to encode binding specificity onto its sequence. Understanding these sulfation patterns constitute a major challenge. Previous studies hinted that sulfate groups can migrate along glycans during collision-induced dissociation in mass spectrometry (MS) experiments, forming isomeric fragments that can lead to incorrect structural assignments. We use ion-mobility – mass spectrometry to investigate the mechanism of this phenomenon in heparin sulfate disaccharides. The sulfate group migrates from the non-reducing to reducing end of the sugar, and the degree of migration does not depend on the structure of the label. The migration product has a sulfate group attached to either 6$O$- or 3$O$-position of GlcNAc, and the migration mechanism consists of multiple steps, with the sulfate group first shifting from the iduronic acid to the 6$O$-position of GlcNAc, and next to the 3$O$-position. The presented data offer insight into the complexity and unpredictability of sulfated sugar fragmentation in tandem MS and extensive investigations is required to determine whether this represents a singular case or a general phenomenon characteristic of deprotonated sulfated glycans.

Isomerisation processes and intramolecular rearrangements, such as double bond shifts or the often-taught McLafferty rearrangement[1], are a common occurrence in classical mass spectrometry (MS) due to energies associated with covalent bond cleavages[2]. Introduction of non-radical ionisation and activation methods promised to decrease the frequency of such events. However, using a combination of tandem MS, infrared ion spectroscopy, and in silico modelling, van Tetering et al.[3] recently demonstrated that intramolecular rearrangements remain a common phenomenon in a gas-phase analysis of molecular ions. Most notably, they called attention to the fact that many entries in MS/MS databases constitute products of iso-merisation—especially cyclisation—of annotated fragments. Particularly, biopolymers are prone to such rearrangements: peptides can undergo sequence-scrambling through a cyclisation mechanism during fragmentation[4–6], and phosphate groups, a common post-translational modification, can migrate along a peptide sequence[7] which leaves the original modification site ambiguous[8–10].

Likewise, in MS-based analysis of glycans, intramolecular rearrangement is a well-known phenomenon. The most prominent example is a fucose migration, but similar rearrangements have been observed for other carbohydrates, such as rhamnose[11], mannose[12], and xylose[13]. In the seminal study from 1995, Kováčik et al.[11] showed that during fragmentation of fucose-containing oligosaccharides, glycans in between the fucose attached close to the non-reducing end and the reducing-end is cleaved, yet the fucose moiety would remain directly attached to the reducing-end glycan. Wuhrer et al.[12] showed that a reducing-end GlcNAc residue can migrate onto the mannose antennas, which leads to the loss of a core GlcNAc unit. Recently, we have demonstrated that the fucose migration occurs during MS-analysis of intact blood-group antigens, and the process requires a mobile proton[14,15]. Therefore, the rearrangement is not induced by the CID-induced fragmentation, but is most likely triggered by the *in-source* activation during the nano-electrospray ionization (nESI). This observation means that the migration can occur at much lower activation energies than expected from a glycosidic bond cleavage. Furthermore, by comparing the cryogenic IR spectra of different antigens with Density Functional Theory (DFT) simulated spectra of potential rearrangement products, we determined the rearrangement product, and that the likeliness of the migration depends on the availability of a mobile proton, which can depend on the glycan conformation determined by its sequence[16]. Despite these efforts, the general migration mechanism has not been fully understood yet[17–20].

[1]Institute of Chemistry and Biochemistry, Freie Universität Berlin, Berlin, Germany. [2]Department of Molecular Physics, Fritz-Haber-Institut der Max-Planck-Gesellschaft, Berlin, Germany. [3]Department of Chemistry, Hunter College, The City University of New York, New York, NY, USA. [4]PhD Programs in Chemistry and Biochemistry, The Graduate Center, The City University of New York, New York, NY, USA. [5]These authors contributed equally: Lukasz Polewski, Murat Yaman. ✉e-mail: mmarians@hunter.cuny.edu; kevin.pagel@fu-berlin.de

Similar to peptides, glycans are frequently modified by other groups such as amine, phosphate, and sulfate groups. Glycosaminoglycans (GAGs), which carry sulfate groups along their sequence, are highly complex linear polysaccharides, often bound to proteoglycans (Fig. 1A), that play crucial roles in biological recognition processes[21–25]. Positional sulfation constitutes the major factor in the structural heterogeneity of GAGs, and the exact position of the sulfate groups is critical for the recognition by and interactions with other proteins[26]. However, these discrete sulfation patterns increase the analytical challenge due to the requirement for sequence-based analysis (Fig. 1B)[27–29]. In 2011, Kenny et al.[30] first reported the CID-induced migration of sulfates along O-linked glycans and GAGs, which can lead to misinterpretations of their biological functions; however, the structure of the rearrangement products, as well as potential mechanism, remains elusive. Herein, we combine MS and IMS experiments with calculations using DFT to investigate the isomerization reaction specific to sulfated glycans in the gas phase, to provide insights into the resulting isomeric products and potential rearrangement mechanisms.

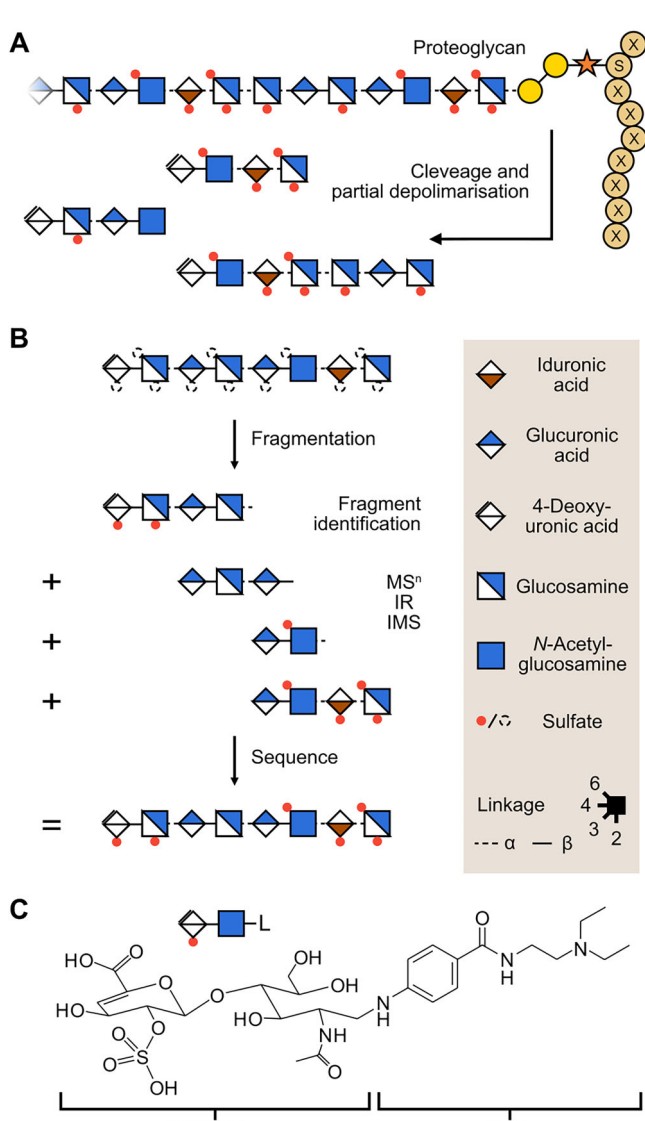

**Fig. 1 | The structure of proteoglycans. A** A heparan sulfate polymer chain is assembled on a protein backbone to form a proteoglycan. **B** The structural analysis of GAG oligosaccharides is usually performed using fragmentation in mass-spectrometry based experiments (left), Symbol nomenclature for glycans (SNFG)[43] (right). **C** Structure of the ProA-labelled HS2SNAc (HS2SNAc-ProA).

## Results

### Sulfate migration in heparan sulfate disaccharides

The CID fragmentation of a singly negatively charged heparan sulfate disaccharide with a procainamide (HS2SNAc-ProA) attached to the reducing end is shown in Fig. 2A. Most of the observed fragments are expected: a $Y_1$ fragment at m/z 439, a cross-ring $^{0,2}x_1$ fragment at m/z 579, and the dehydrated fragments at m/z 421 and 561, respectively. However, a surprisingly abundant but unexpected fragment is detected at m/z 519. The mass difference between this unknown peak and the $Y_1$ fragment is 80 Da, which indicates a sulfate modification ($+SO_3$). However, the sulfate of the used standard is linked to the 2O-position of the uronic acid, and a sulfated $Y_1$ fragment, consisting of GlcNAc and the label, should not be formed. As a result, the sulfate group must change the position from the uronic acid to the adjacent GlcNAc after the in-source ionization (Fig. S1) but prior to the fragmentation of the molecule.

There are several possible acceptors of the sulfate group on the $Y_1$ ion: hydroxy groups, the amide function of the GlcNAc, and the functional groups of the label procainamide. To unravel the identity of the unknown ion, additional ion mobility- mass spectrometry (IM-MS) experiments of the fragments were performed. The arrival time distributions (ATDs) of the $Y_1 + SO_3$ fragment revealed two resolvable peaks with roughly equal intensity (Fig. 2B, red). To determine whether these peaks belong to two or more isomeric products of the rearrangement or a single isomer that forms multiple conformers, two GlcNAc-ProA standards sulfated at the 6O- or 3O-position were synthesised (see methods for details). Both synthetic standards yield a single peak in the ATDs, and a comparison to the $Y_1 + SO_3$ fragment of HS2SNAc-ProA shows a very good agreement between the respective arrival times. Specifically, GlcNAc-6S-ProA matches the more compact $Y_1 + SO_3$ fragment with a drift time of ~4.4 ms (Fig. 2B, yellow, $^{DT}CCS_{He}$ of 151 Å²), while GlcNAc-3S-ProA fits the more expanded structure that corresponds to the ATD peak at ~5.6 ms (Fig. 2B, blue, $^{DT}CCS_{He}$ of 168 Å²).

### Structure of migration products

To validate this assignment and to exclude contribution of the other structural isomers that carry a sulfate group at position 4O or 5O, we have turned to methods of computational chemistry. Previously, the semi-empirical and DFT-based methods have been successful in the prediction of gas-phase structures of glycans[17], glycoconjugates[31], RNA rearrangement products[32], and amino acid complexes[33]. A detailed protocol of the employed computational routine is described in the methods. Briefly, a conformation sampling of the $Y_1$ ion with the sulfate group at positions 3O (GlcNAc-3S-ProA) and 6O (GlcNAc-6S-ProA) was performed using iterative metadynamics combined with the genetic crossing (iMTD-GC) algorithm implemented in CREST[34]. The calculations have been performed using GFN2-xTB level of theory, which is generally accurate for carbohydrates[35,36]. However, we observed that the resulting conformational ensembles have been biased towards compact structures (Fig. S2); to expand the conformational search, we performed a separate set of ab initio molecular dynamics (MD) simulations using same level of theory at 400 K and 450 K (Fig. S3). Next, we subsampled the molecular trajectories for structures of different sizes and added them to the analysed structures. All geometries were then optimized using PBE0-D3/6-311 + G(d,p) level of theory, and their relative free energies, calculated using harmonic approximation at 300 K, were plotted against the calculated $^{TM}CCS_{He}$ values computed with a Trajectory Method available in MobCal (Fig. 2C)[37].

The resulting conformational hierarchies reveal that the free-energy surfaces of the two ions have a very distinct shape. Conformations of the GlcNAc-6S-ProA ion form a rather narrow funnel around the most stable and compact conformer ($^{TM}CCS_{He}$ of 152 Å²). Therein, the negatively charged sulfate group on the 6O position of GlcNAc is stabilized by three H-bonds: one with the O5H, one with N-acetyl group of GlcNAc, and another N-acetyl, which is a part of the ProA label. This last interaction causes the ProA label to fold back to the glycan chain to reach the sulfate group, which results in a compact conformation of the isomer. In contrast, the GlcNAc-

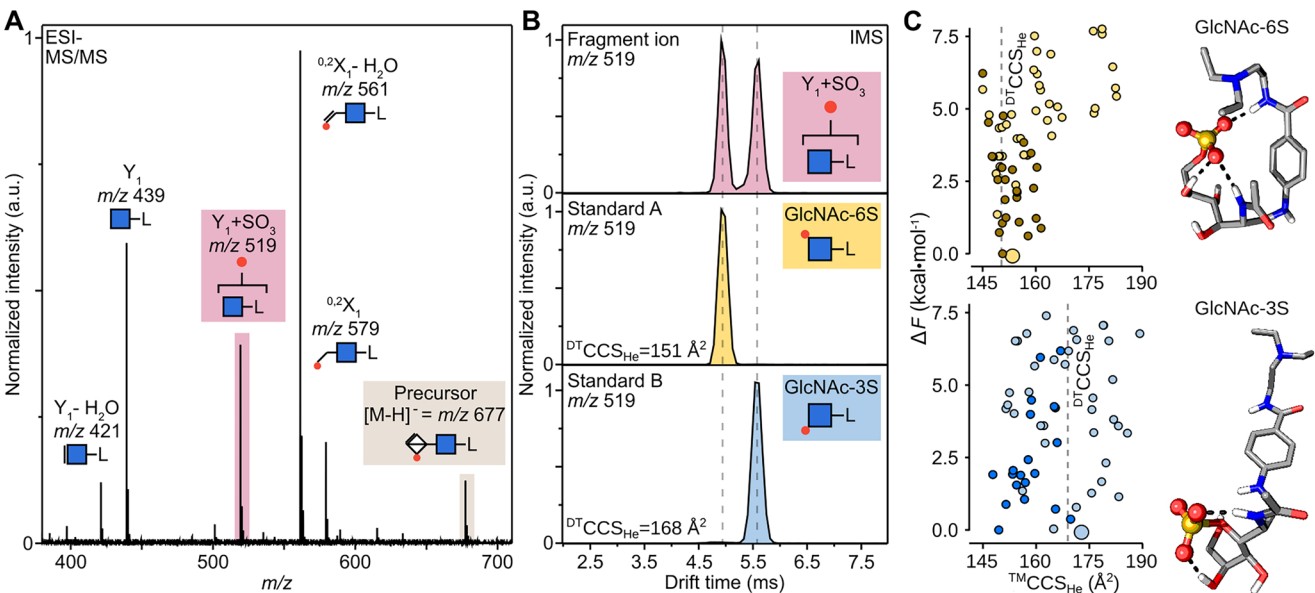

**Fig. 2 | Fragmentation of HS2SNAc-ProA. A** CID-MS/MS of HS2SNAc-ProA. Besides expected fragment ions (grey), an unexpected $Y1 + SO_3$ ion (red) is formed through sulfate migration. **B** Arrival time distribution (ATDs) of the $Y_1 + SO_3$ fragment ion (red) and two standards, GlcNAc-6S-ProA (yellow) and GlcNAc-3S-ProA (blue), and the respective $^{DT}CCS_{He}$ derived for both standards. **C** Relative free energy of GlcNAc-6S-ProA (yellow) and GlcNAc-3S-ProA (blue) conformers against their simulated collision cross-section ($^{TM}CCS_{He}$). Dark and light shades indicate two different conformational sampling techniques, and the dashed line indicates experimental CCS. The structure of the most stable conformer, marked with a larger circle, is shown next to the respective plot. The grey dashed line represents the experimental CCS ($^{DT}CCS_{He}$).

3S-ProA ion shows a glass-like free-energy surface with multiple equally stable minima of different CCSs. This conformational flexibility arises from the sulfate group being at $3O$ where it is able to form H-bonds with adjacent $5OH$, $6OH$, and $N$-acetyl group of GlcNAc, as well as the amine and amide functional groups on the ProA label. These interactions collectively result in a more diversified hydrogen bond network, which leads to stable conformers of different CCSs, with the most stable conformer having $^{TM}CCS_{He}$ of 173 Å². In effect, the difference of 21 Å² between the most stable conformers of the two isomers agrees with the 17 Å² difference between GlcNAc-6S-ProA (compact ion) and GlcNAc-3S-ProA (extended ion) observed in IM-MS experiments.

DFT was also used to investigate whether the sulfate can migrate to the positions $4O$ or $5O$ of the GlcNAc for which synthetic standards were not available. The conformational search followed a similar procedure to GlcNAc-3S-ProA and GlcNAc-6S-ProA ions (Fig. S4). For GlcNAc-4S-ProA, the predicted $^{TM}CCS_{He}$ of the three most stable conformers, all within 1 kcal mol$^{-1}$, were found to be 148 Å², 156 Å², and 175 Å². Similarly, for the $5O$-position ion, the most stable conformer has a $^{TM}CCS_{He}$ value of 144 Å², followed closely by two other conformations at 149 Å² (less stable by 0.5 kcal mol$^{-1}$) and 173 Å² (less stable by 1.2 kcal mol$^{-1}$). These results indicate that both hypothetical ions should exhibit a rather large conformational freedom, similar to that of GlcNAc-3S-ProA. Comparison of the relative free energies of four different isomers, which previously helped to identify the products of fucose migration in Lewis antigens, shows less than 1.0 kcal mol$^{-1}$ difference in stability between the four isomers (Fig. S2), and thus we cannot fully dismiss the formation of either of these two additional isomers at this stage.

**Label is not responsible for inducing the sulfate migration**
Next, we examined impact of the label on the occurrence the sulfate migration and the product being formed. To do so, we appended HS2SNAc with three additional reducing-end labels: benzocainamide, 4-aminobenzoic amide, and aniline. These labels were chosen to gradually reduce the chemical complexity and size of the attachment to pinpoint structural elements that trigger the migration: benzocainamide is missing the tertiary amine; 4-aminobenzoic amide lacks the ethyl chain, and lastly, aniline, which is missing the $N$-acetyl group. The CID-fragmentation spectra of these labels at identical instrumental conditions, the formation of the $Y_1 + SO_3$ migration product remained comparable between different labels (Fig. 3A). The slight changes in intensity, which are also visible in the relative precursor intensities, can be assigned to the mass-dependent energy transfer efficiency during the CID process. Hence, the degree of formation of the sulfated $Y_1$- fragment is not dependent on the type of reducing-end label.

Next, we examined possible influence of the label on the resulting migration site (Fig. 3B, C). The benzocainamide labelled species showed a very similar ATD to the ProA labelled disaccharide. The peaks appear less resolved and are shifted towards lower drift times due to mass differences between the labels, but qualitatively match the previous results of two ions of different shapes. Merely a shoulder is visible in the ATD of the $Y_1 + SO_3$ migration product of the structures that are labelled with 4-aminobenzoic amide. Lastly, the IMS result for the aniline-labelled structures shows a single peak. CCS values of $Y_1 + SO_3$ structures with three additional labels predicted from DFT-derived conformations agree that the difference in arrival times between the two isomers should decrease with the reduction of the label functionality. The comparison of the $^{TM}CCS_{He}$ of the most stable conformers of each isomer (Fig. 3C) shows that the difference of 21 Å² between two ions labelled by ProA decreases to 18, 17, and 7 Å² for the benzocainamide-, 4-aminobenzoic amide-, and aniline-labelled structures, respectively. While the number of hydrogen bonds stabilizing the sulfate group in both isomers of all three labels remains the same as ProA-labelled structures, the smaller label size decreases the CCS difference between the labels. The utilized label, therefore, likely does not influence the migration process, neither quantitative nor qualitative, but can be crucial for differentiation of migration sites.

**Sulfate migration mechanism**
Finally, we attempted to deduce the order of formation of the two $Y_1 + SO_3$ fragments using energy-resolved IM-MS experiments. To accomplish it, we monitored the ATD of the $Y_1 + SO_3$ fragment by step-wise increasing collision voltages in the CID cell (see methods for the details). Surprisingly, the two migration products form at slightly different rates. The formation of GlcNAc-3S-ProA occurs preferably at a higher collision voltage compared

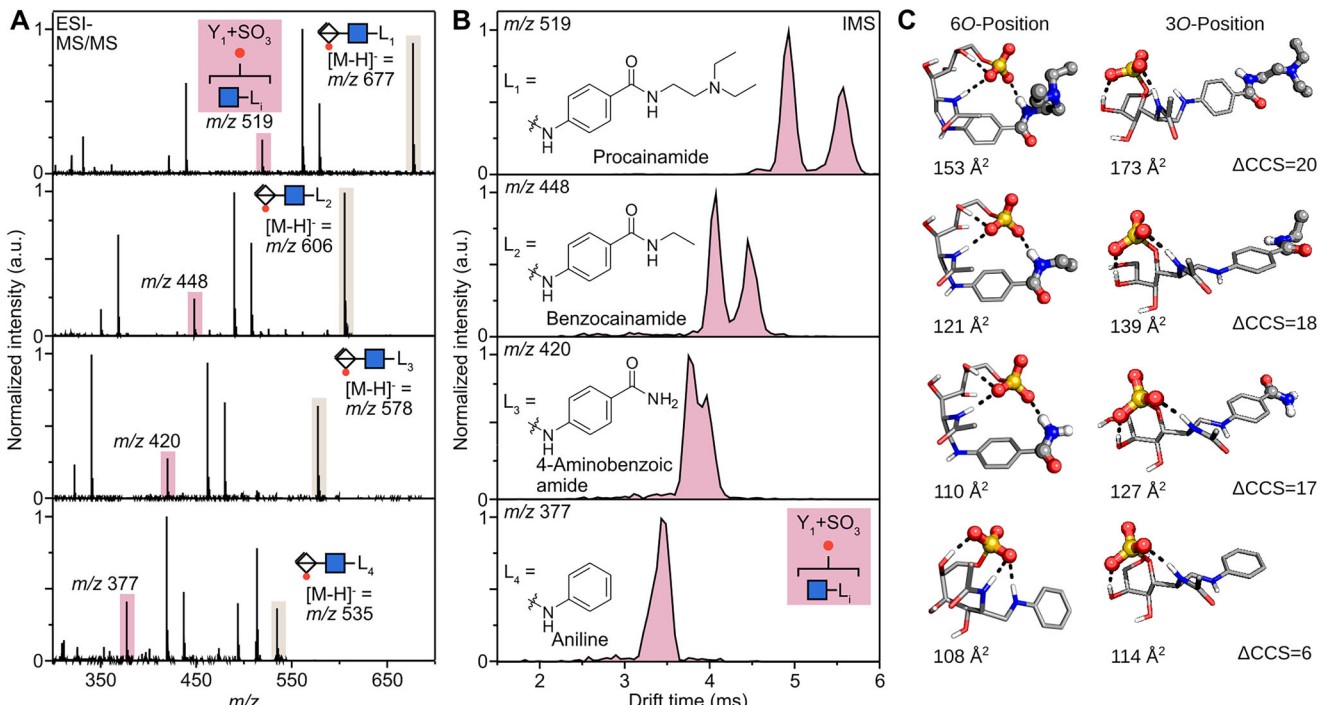

**Fig. 3 | Fragmentation of HS2SNAc labelled with three alternative reducing-end labels. A** CID-MS/MS spectra of benzocainamide, 4-aminobenzoic amide, and aniline labelled HS2SNAc. The extent of $Y_1 + SO_3$ fragment formation is similar for all labels. **B** ATDs of the $Y_1 + SO_3$ fragments. Sulfate migration appears to occur throughout; the CCS difference is reduced with decreasing size of the label. **C** Predicted lowest energy conformers of $Y_1 + SO_3$ fragments with different labels, their $^{TM}CCS_{He}$ values, and the difference between CCSs of the two fragments ($\Delta CCS$).

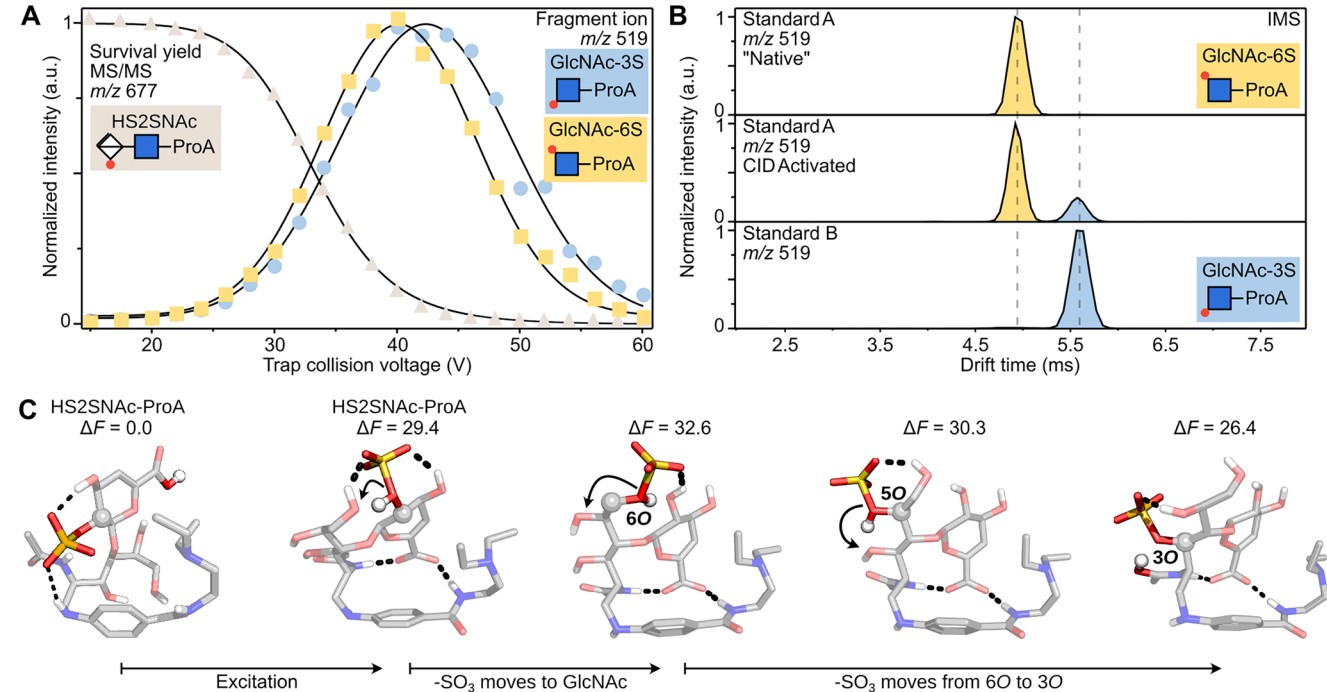

**Fig. 4 | Energy resolved CID measurements of HS2SNAc-ProA. A** Survival yield curve of HS2SNAc-ProA and two fragment ions, highlighting the delay. The intensity drop of the fragments at higher collision voltage is caused by further fragmentation of **B** ATDs of GlcNAc-6S-ProA and GlcNAc-3S-ProA. Activation of the GlcNAc-6S-ProA standard leads to the formation of GlcNAc-3S-ProA. **C** Proposed steps of the sulfate migration leading to GlcNAc-6S-ProA and GlcNAc-3S-ProA products. The free energy values are given in kcal mol$^{-1}$.

to the formation of GlcNAc-6S-ProA. This is especially clear when the survival yield curves are compared (Fig. 4A). At a collision voltage of 40 V the relative intensity of GlcNAc-6S-ProA reaches its maximum. Compared to that, GlcNAc-3S-ProA reaches its maximum at 44 V, which leads to an inversion of the relative intensities of the two migration products at 42 V, with more GlcNAc-3S-ProA being formed than GlcNAc-6S-ProA. This effect could be caused either by a difference in fragment formation or by a difference in fragment stability in the gas phase, particularly considering the

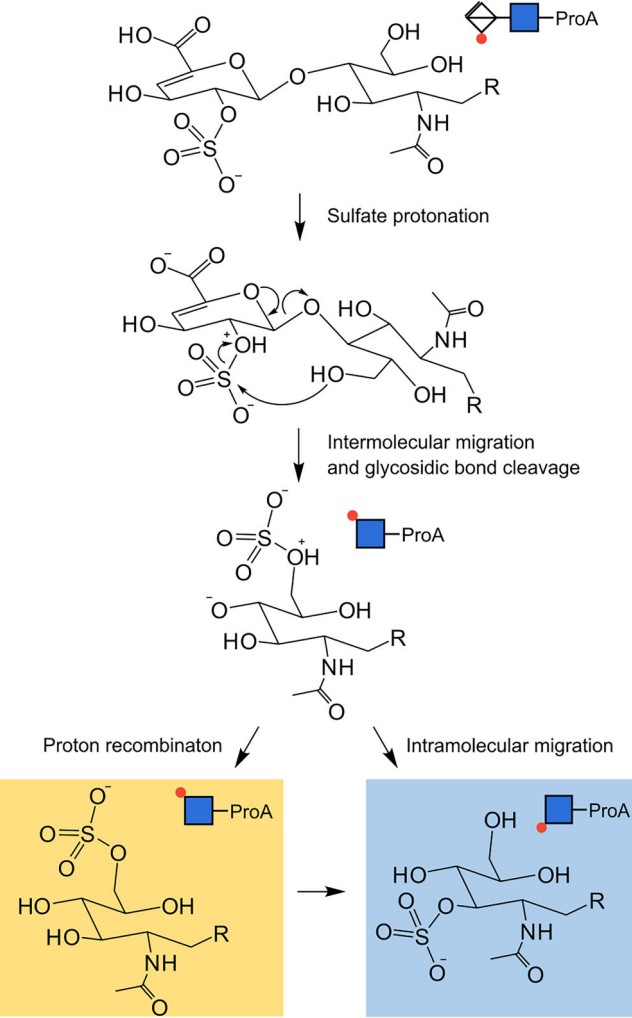

**Fig. 5 | The proposed sulfate migration mechanism of heparan sulfate (HS2SNAc-ProA).** First, the mobile proton shifts to the sulfate group, activating it for migration to adjacent O6. Sulfate migration and glyosidic bond cleavage occur in presumably concerted step. The resulting reactive fragment can either recombine to form GlcNAc-6S-ProA or continue with sulfate migration via O5 until it reaches the most stable position at O3 in GlcNAc-3S-ProA.

number of hydrogen bonds that stabilize the sulfate groups. The two synthetic standards show near identical survival yields (SY50) with 36.3 V and 36.6 V for the GlcNAc-3S-ProA and GlcNAc-6S-ProA compounds (Fig. S5). Therefore, it can be concluded that the formation of GlcNAc-3S-ProA is preferred at higher collision voltages.

While HS2SNAc-ProA shows no signs of migration prior to fragmentation (Fig. S6), the step-wise nature of the migration process, which occurs due to negative charge moving between hydroxyl groups, can be confirmed by monitoring the drift time of the GlcNAc-6S-ProA standard during the CID activation (Fig. 4B). The drift time of the migrated product matches the drift time of the GlcNAc-3S-ProA standard, confirming the migration site. The overall extent of migration is CID voltage-dependent, with higher voltages leading to more relative migration (Fig. S7), and the lack of other bands in any of these experiments suggests that GlcNAc-4S-ProA and GlcNAc-5S-ProA isomers are not present, in agreement with earlier predictions by DFT calculations.

To further elucidate why GlcNAc-6S-ProA forms at less harsh conditions than GlcNAc-3S-ProA and propose a sulfate migration mechanism, we turned to DFT calculations again. First, to account for molecular excitation, we investigated structures of two protomers of the HS2SNAc-ProA: one with protonated carboxylic acid group, and one when the proton is

shifted to the sulfate group. Conformational sampling showed that the proton shift changes the structure of the most stable conformers. When the carboxylic group is deprotonated, it rotates to interact with the ProA label, which positions the protonated $SO_3$ group next to the $6O$ on GlcNAc. In this conformation, the two isomers with the sulfate group at 4-deoxy iduronic acid and at $6O$-position of the GlcNAc residue differ only by 3.2 kcal mol$^{-1}$ (Fig. 4C), and proximity of the two positions can facilitate the $SO_3$ migration. After this initial shift, the sulfate group can rearrange further to more stable positions at $5O$ and $3O$ of the GlcNAc residue. The presented mechanism has been summarized in Fig. 5. Briefly, the rearrangement relies on the presence of an acidic/mobile proton and involves a proton transfer to the sulfate group, which changes the conformation of the ion. The activated sulfate then moves to the adjacent $6O$-position of the neighbouring glycan. Since HS2SNAc-ProA shows no signs of migration prior to $Y_1$-fragment formation, a concerted mechanism is assumed, where sulfate migration and glycosidic bond breakage happen simultaneously. Therefore, in the first step, glycosidic bond fragmentation and sulfate migration lead to the formation of GlcNAc-6S-ProA in its active, protonated form. In the second and final step, sulfate migration to the $3O$ position occurs.

## Discussion

Previous studies provided only indirect evidence for sulfate migration during glycan CID experiments[30]. Here, we present the first detailed mechanistic investigation of this phenomenon using a set of small, derivatized glycosaminoglycan oligosaccharides. In CID fragmentation of HS2SNAc-ProA, we observed sulfate migration from the terminal to the reducing-end sugar. Using ion mobility-mass spectrometry (IM-MS) and DFT calculations, we identified the migration site by comparison with synthetic standards. We also examined how different reducing-end labels affect the migration products. Energy-resolved IM-MS experiments and DFT calculations revealed that the migration proceeds through multiple steps, with sulfate shifting first to the $6O$-position and then to the $3O$-position.

Although only a limited set of structures was studied, the findings have significant implications for the structural analysis of sulfated glycans by tandem mass spectrometry. As shown here, sulfate migration requires less activation energy than bond cleavage, making it possible for a sulfate group to relocate within the molecule without detection. These rearrangements produce isomeric fragments that are indistinguishable from the original structure unless additional techniques such as IM-MS are employed. This can lead to misinterpretation and incorrect structural assignments in tandem MS data. An open question remains as to how broadly this migration occurs. While our current data suggest that the influence of reducing-end labels is minor, further studies are needed to assess the role of adducts, charge polarity, and precursor structure in promoting sulfate migration.

## Methods
### Chemicals
All chemicals and solvents were purchased from Sigma-Aldrich (St. Louis, USA) and used without further purification. The HS2SNAc standard was purchased from Iduron (Cheshire, UK). HPLC-grade solvents were used throughout. GlcNAc-3S was synthesized by acetylation of GlcN-3S according to a method described previously[38]. Reducing-end labelled glycans were synthesised according to established protocols (as per Ludger ProA labelling kit)[39,40].

### MS and IM-MS measurements
For MS analysis, samples were dissolved prior to use with water:methanol:ammonium acetate (1:1:50 v:v:mM) to yield 5–10 μM analyte solutions. Measurements were performed on a Waters Synapt G2-S using direct-injection nESI. A capillary voltage of 0.8 kV was used, with a source temperature of 150 °C. Standard TW-IMS settings were used with a trap wave height of 40 V, velocity of 311 m/s, IMS wave height of 40 V, velocity of 450 m/s, and transfer wave height of 4 V, velocity of 175 m/s. Energy-

resolved CID experiments were measured in steps of 2 V in the trap cell, using nitrogen as collision gas. $^{DT}CCS_{He}$ values were measured on a modified Synapt G2-S system with the step-field method, utilizing helium as a drift gas[41]. The notation follows the community guidelines on the reporting of CCSs[42].

## Conformation search

In this study, the conformation searches were carried out within two different methods in order to expand the space of molecular conformations. First, we performed iterative metadynamics combined with the genetic crossing (iMTD-GC) algorithm implemented in CREST[34], using default settings to identify the lowest-energy conformers with the limitation of the energy window of 5 kcal mol$^{-1}$. To expand the conformational search further, the ab initio molecular dynamics (AIMD) simulations using xTB package were employed at the same GFN2-xTB level of theory. The simulations were performed at two temperatures, 400 and 450 K, for 1 ns and using 1 fs timestep. The same procedure was applied to all glycans with four different labels.

## Collision cross section calculations

The collision cross section calculations were performed using the trajectory method (TM) available in the MobCal-MPI 2.0[31,32,37] software. To perform CCS calculations using MobCal 2.0, we computed atomic charges using CHELPG partition scheme available in Gaussian16 using recommended ωB97X-D3/def2-TZVPP level of theory. Each $^{TM}CCS_{He}$ value was calculated in helium as buffer gas at 298 K, employing van der Waals parameters provided by the software, based on the structures generated by conformational search methods, CREST, and AIMD. Furthermore, structures from 400 and 450 K were combined and systematically selected based on the criteria of every 1 Å$^2$ increasing from the minimum to the maximum CCS values.

## Density functional theory (DFT) calculations in the gas phase

The conformers generated by conformational searches with CREST and AIMD were reoptimized using the Perdew, Burke, and Ernzerhof (PBE0) hybrid functional augmented with GD3BJ empirical dispersion force corrections (PBE0 + D3/6–31 + (d,p)) in the gas phase in Gaussian16 software. The structures were then clustered to remove duplicates, and the structures within 5 kcal mol$^{-1}$ energy window above lowest energy conformer were reoptimized in a larger basis set, 6–311 + (d,p), followed by the determination of vibrational frequencies via harmonic approximation.

## Data availability

The authors declare that the data supporting the findings of this study are available within the paper and its supplementary information files.

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

## Acknowledgements

K.P. and L.P. thank the Deutsche Forschungsgemeinschaft (DFG, German Research Foundation) for financial support of this work under the CRC 1340 "Matrix in Vision" grant number 372486779. K.P. further acknowledges funding via the European Union's Horizon 2020 Research and Innovation Programme grant number 899687-HS-SEQ. Research infrastructure was provided by the research building SupraFAB realized with funds from the Federal Government and the city of Berlin.

## Author contributions

K.P. and M.M conceived this project. L.P. and M.T. performed the experiments, and M.Y. performed computational analysis. All authors corresponded equally to writing this manuscript.

## Funding

## Competing interests

The authors declare no competing interests.
