## [Transparent Peer Review file · Communications Chemistry]

Mechanistic Study of Sulfate Migration in Glycosaminoglycans during MS Fragmentation

Corresponding Author: Professor Kevin Pagel

This manuscript has been previously reviewed at another journal. This document only contains information relating to versions considered at Communications Chemistry.

Version 0:

Reviewer comments:

Reviewer #1

(Remarks to the Author)

Reviewer 2 writing review for Reviewer 1 and 2 to be best of my ability.

We thank the referee for their time and insight with providing comments to help us develop a stronger manuscript. We have addressed the comments and provide a detailed response below.

Both reviewers commented that “the results and mechanisms have been reported previously” and that it “is a well-known phenomenon in MS studies and has been studied extensively by other labs.” We strongly disagree with this statement and kindly request the reviewers to substantiate their claims with citations.

The only report in which sulfate migration was described previously is cited and thoroughly discussed in the manuscript (reference 20 in the manuscript, Kenny, D. T. et al.). This work describes the phenomenon and discusses the impact of a mobile proton on the migration process. However, it was not possible to identify the migration product and propose a mechanism. The present work is therefore not a simple repetition; it provides the first mechanistic insight into this crucial rearrangement reaction which has fundamental consequences for the MS-based analysis of glycosaminoglycans.

My (Reviewer 2) statement was and still is in regards to a general incident of structure rearrangement, which is commonly observed and addressed. Yes, the mechanism of sulfate migration is not well understood or elucidated. However, this is only a small novelty niche, which I believed is a major reason this MS did not warrant the original Nature Communication publication. From my understanding, a research has to demonstrate highly novel and groundbreaking science to justify publication by Nature Communication. Again, this is my subjective opinion.

Reviewer #1 (Remarks to the Author):

The authors investigated sulfate migration using IMS, DFT, and synthetic chemistry. Negative-mode ESI was used to generate the deprotonated parent ion. Sulfate migration was still observed due to the presence of a proton on the sulfate group. Four different tags were used to study the effect of labeling. This is a decent study; however, the results and mechanisms have been reported previously. There are several points the authors may need to consider.

1. On page 5, the authors stated that “As a result, the sulfate group must change its position from the uronic acid to the adjacent GlcNAc prior to fragmentation of the molecule.” There are two possibilities.

First, the migration may occur during ESI, as the authors stated in the manuscript. However, the authors need to vary ESI parameters (such as the vaporization temperature and tube lens voltage) to verify this. Second, the migration may occur during activation in the gas phase. I personally think the latter is the case.

A variation of the ESI parameters would lead to activation prior to the first quadrupole, i.e. in-source activation. In the utilized instrument and given the small size of the studied system, this type of activation only marginally differs from the trap activation shown here in Figures 4, S5, S6 and S7. However, here trap activation experiments after quadrupole isolation of the precursor were performed, which provides much more controlled and comparable conditions. Following migration after in-source activation on the other hand, would lead to ambiguous data that cannot be interpreted.

The trap activation method makes sense since this minimizes spread in internal energy of ions.

In order to clarify that there is no migration prior to CID, we have added Figure S1 which shows the intact HS2SNAc (the precursor) peak in comparison to a HS6SNAc (the migration product) standard; identical source parameters have been used for this measurement and the following ones, hence in this case, no migration prior CID activation is occurring. To reflect this comment, we have adjusted the sentence on page 5 from:

“As a result, the sulfate group must change its position from the uronic acid to the adjacent GlcNAc prior to the fragmentation of the molecule.”

to

“As a result, the sulfate group must change its position from the uronic acid to the adjacent GlcNAc after the in-source ionization (Figure S1) but prior to the fragmentation of the molecule.”

This is an acceptable response.

2. To further investigate the mechanism, the authors may consider protecting acidic sites to ensure there is no mobile proton. We thank the reviewer for this comment. The necessity of the mobile proton was discussed extensively in the previous work by Kenny et al. (reference 20 in the manuscript, Kenny, D. T.; Issa, S. M. A.; Karlsson, N. G., Sulfate migration in oligosaccharides induced by negative ion mode ion trap collision-induced dissociation. *Rapid Communications in Mass Spectrometry* 2011, 25 (18), 2611-2618.) A repetition of these experiments would not lead to further insights.

Overall, this is a good study, but it may not be novel enough to be published in *Nature Communications*.

This is a good suggestion by Reviewer 1 since it can help rule out other mechanisms such as the vehicle mechanism which relies on diffusion of the sulfate group. However, since the mechanism is stated as concerted by the author then it may be more Grotthus-like.

Reviewer #2 (Remarks to the Author):

Glycan sulfation has important biological functions such as protein binding specificity, structural integrity, and cell signaling. It may therefore be crucial and helpful to determine the native atomic site of sulfation to potentially advance biomarker discovery, diagnostics, or therapeutic design. Elucidation of a glycosaminoglycan sulfation site(s), for example, can be achieved through fragmentation-based sequencing using collision induced dissociation (CID) and traditional mass spectrometry. However, sulfate migration has been observed to occur during the CID process, which makes structural analysis very challenging since traditional MS cannot properly separate and detect isobars. Indeed, structural rearrangement, as noted by the authors, is a well-known phenomenon in MS studies and has been studied extensively by other labs for a range of chemical systems. Hence, the overall novelty of the work is limited. In this work, Polewski et al. aim to use ion mobility mass spectrometry (IM-MS) and molecular modeling to elucidate both the isomeric products (i.e., isobars) and the inherent mechanisms of sulfate group rearrangement. Although this work explores an important and interesting phenomenon in the mass spectrometry domain, the conclusion relies on limited data. Additionally, the computational methods need further clarification.

A large number of papers have appeared on CCS modeling and the use of ML tools to compute CCS values. However, none of this is noted – even as a justification for their computational protocol, which is not novel in light of earlier work. We agree with the reviewer that the ML tools become useful for the prediction of CCS values. Nonetheless, the training and parametrization of such moments requires rigorous validations, particularly for structurally and conformationally complex molecules such as GAGs, and such work is outside of this study. Furthermore, the trajectory method used in this study, which has been implemented in MobCal 2.0 in 2023 (Haack, A. et al.; *Analyst* 2023, 148, 3257–3273) has is still under active development, offers a well-tested and cost-effective solution to determination of the CCSs of small molecules (recently: Gorbachev et al, *JACS*, 2025, 147, 4308). Finally, the computational cost of determining atomic charges for the potentials used in trajectory methods is marginal when compared to conformational sampling at the DFT level of theory, which reduces the need for ML-based approaches.

The concern here that remains unaddressed is that a lot of CCS modeling works have been done but not enough are noted by the authors. It is understandable that there may not have been a justification for using the software/computational tools and that the method is an arbitrary choice. However, it should be noted that different results may be produced from different sampling method and/or different DFT level of theory. This is something the authors should emphasize in the MS since the test systems are very few.

Overall, rearrangement of molecular ions is a well-known problem, the experimental methods used are routine and the modeling uses approaches that have been well described previously. Given all this the paper is not well suited for communications oriented journal like *Nature Communications* and would be better suited for *JASMS* or a related MS focused journal.

We respectfully disagree with the reviewer on this point. Migration reactions have indeed been reported for phosphate groups in peptides and fucose residues in glycans. However, as mentioned above there is only a single report that describes the rearrangement of sulfates. This report is cited and discussed in the manuscript.

The sulfation pattern of glycosaminoglycans (GAGs) – often referred to as the “sulfation code” – is a defining feature of their biological function. Deciphering this code is inherently challenging and relies almost entirely on mass spectrometry and tandem MS-based techniques. As such, the findings we present here have important implications for the structural and functional analysis of GAGs. Given the increasing interest in this area, further accelerated by developments in infection biology during the COVID-19 pandemic, we believe these insights are timely and highly relevant to the broader glycobiology community.

As I stated above, rearrangement in a general sense is a commonly observed and discussed problem for molecular ions in mass spectrometry.

Beyond these issues the following concerns should be addressed.

Comments:

1. The experimental CCS values for GlcNAc-6S and GlcNAc-3s were obtained from the arrival time distributions (ATD) of their respective ions. Typically, the ATD is influenced by the ions' conformational heterogeneity. With that said, why was the calculated (computationally produced) CCS value per system not Boltzmann-weighted with respect to the ensemble relative energies to simulate a CCS that accounts for conformation heterogeneity.

We fully agree with the reviewers that, assuming that the relevant parts of the molecular phase space is ergodic, the ATDs

represents conformational average of the respective ions. However, the accurate determination of such Boltzmann-weighted CCSs would require either averaging over full conformational space or accurate calculations of the free energy of most important conformational states. Histograms of multiple 1 ns MD simulations (GNF2-xTB level of theory) are presented in the SI, but neither such trajectories represent a fully converged sampling, nor the level of theory offers accurate representation of the PES. On the other hand, averaging over most stable minima on the PES derived at higher level of theory (DFT) suffers from the shortcomings of the harmonic approximation: it underestimates the population of extended conformers. Thus, while we report the absolute values of the most stable (and relevant conformers), our analysis focuses more on changes to the CCSs between different isomers and labels.

I believe that authors can just Boltzmann-weighted average over DFT electronic energies obtained for the ensemble of conformations optimized. These are the same conformations used to calculate the CCS values, if I am not mistaken. From my understanding, an ATD curve shape is influenced by different conformers or isomers populated. I am curious to see what the weighted average CCS is when conformers and isomers are all taken into account.

2. The two sampling techniques used for structure assignments favor (indicated as a bias in the manuscript) either GlcNAc-6S or GlcNAc-3s (Figure 2C). Were two sampling techniques implemented from the start of this work or was it added after to improve the agreement with experiment? Since the purpose of using two different sampling techniques is to sample a broader conformational space, why not use a conformation generator with no energetic preference. This approach would remove sampling bias.

In principle, conformational sampling without use of energy function as a bias means constitutes a systematic or a random search. However, these two classical approaches are well known to fail when a large number of degrees of freedom becomes a factor. In the context of this work, we decided to combine two conformational techniques to explore larger conformational space which is not uncommon in complex molecular systems. For example of such approach, please see Scutelnik et al, *J. Am. Chem. Soc.* 2018, 140, 24, 7554–7560.

The success in capture IM-MS relevant ions is improved with an increase in the number of conformations sampled

3. To help with reproducibility, please indicate which charge scheme was used in Gaussian16 to obtain atomic partial charges for the CCS calculation using the trajectory method.

The protocol to calculate the charges is clearly explained in the original publication (Haack, A. et al.; *Analyst* 2023, 148, 3257–3273). We have clarified in the methods and SI sections that we have used the same partition scheme and level of theory to compute the atomic charges:

“To perform CCS calculations using MobCal 2.0, we computed atomic charges using CHELPG partition scheme available in Gaussian16 using recommended ω B97X-D3/def2-TZVPP level of theory.”

This is an adequate response.

4. Line 322 in the manuscript states that TW-IMS (Traveling Wave Ion Mobility Spectrometry?) settings were used, however, throughout the text the DT superscript is attached to CCS (i.e., DTCCSHe). If not mistaken, DT is for drift tube ion mobility spectrometry. Thus, were the experimental CCS values obtained from TWIMS or DTIMS?

The sub- and superscripts are descriptors of the measurements conditions and follow established community recommendations. For details see doi: 10.1002/mas.21585.

Measurements were performed on both, a standard TW Synapt G2-S and a modified DT Synapt G2S. The presented ATDs are from TW-IMS measurements in N₂ as these are usually better resolved; the reported DTCCSHe values were derived from direct CCS measurements on a DT instrument using the stepped field method. Helium was chosen as a drift gas as it can be described more accurately using theoretical calculations. For details see doi: 10.1002/mas.21585.

So the DTIMS data were used as calibration standards for assigning CCS values to the TWIMS ATD? If this is true then the authors should make this more clear for those readers who are not knowledgeable in mass spectrometry procedures/technologies.

5. Are the conformation sampling temperatures of 400K and 450K similar to the temperatures that were experimentally observe during the CID event or drift region? Or were these temperatures selected arbitrarily to enhance s conformational sampling?

In the present case, measurements were performed at ambient conditions using room temperature drift gas. However, the effective temperature of the ions is certainly higher, especially after CID. Measuring this effective temperature is non-trivial and beyond the scope of this work. In order to still sample a reasonable range of conditions, the temperatures were arbitrarily chosen.

This is a reasonable response.

Reviewer 2 writing reviews for Reviewer 1 and 2 to be best of my ability.

We thank the referee for their time and insight with providing comments to help us develop a stronger manuscript. We have addressed the comments and provide a detailed response below.

Both reviewers commented that “the results and mechanisms have been reported previously” and that it “is a wellknown phenomenon in MS studies and has been studied extensively by other labs.” We strongly disagree with this statement and kindly request the reviewers to substantiate their claims with citations.

The only report in which sulfate migration was described previously is cited and thoroughly discussed in the manuscript (reference 20 in the manuscript, Kenny, D. T. et al.). This work describes the phenomenon and discusses the impact of a mobile proton on the migration process. However, it was not possible to identify the migration product and propose a mechanism. The present work is therefore not a simple repetition; it provides the first mechanistic insight into this crucial rearrangement reaction which has fundamental consequences for the MS-based analysis of glycosaminoglycans.

My (Reviewer 2) statement was and still is in regards to a general incident of structure rearrangement, which is commonly observed and addressed. Yes, the mechanism of sulfate migration is not well understood or elucidated. However, this is only a small novelty niche, which I believed is a major reason this MS did not warrant the original Nature Communication publication. From my understanding, a research has to demonstrate highly novel and groundbreaking science to justify publication by Nature Communication. Again, this is my subjective opinion.

We still disagree with this assessment. A well-known and extensively studied rearrangement reaction in glycans is fucose migration, in which a monosaccharide residue is cleaved from the glycan and subsequently reattached at a different position, often requiring surprisingly little activation. This process is highly relevant for the structural analysis of N- and O-glycans. In contrast, the rearrangement process reported here involves the migration of negatively charged sulfate groups, which exhibit fundamentally different chemical properties. Sulfate groups play a central role in glycosaminoglycans (GAGs), an entirely distinct class of glycoconjugates that requires different analytical strategies and poses different structural challenges.

The only similarity between the two systems is that both involve a rearrangement within a glycoconjugate. Beyond this very general aspect, the processes are not comparable. Glycoconjugates represent the most abundant class of biomacromolecules on Earth and encompass systems as diverse as plant polysaccharides, mucins and human milk oligosaccharides. Drawing parallels based solely on this broad classification is inadequate.

Reviewer #1 (Remarks to the Author):

The authors investigated sulfate migration using IMS, DFT, and synthetic chemistry. Negative-mode ESI was used to generate the deprotonated parent ion. Sulfate migration was still observed due to the presence of a proton on the sulfate group. Four different tags were used to study the effect of labeling. This is a decent study; however, the results and mechanisms have been reported previously. There are several points the authors may need to consider.

1. On page 5, the authors stated that “As a result, the sulfate group must change its position from the uronic acid to the adjacent GlcNAc prior to fragmentation of the molecule.” There are two possibilities.

First, the migration may occur during ESI, as the authors stated in the manuscript. However, the authors need to vary ESI parameters (such as the vaporization temperature and tube lens voltage) to verify this. Second, the migration may occur during activation in the gas phase. I personally think the latter is the case.

A variation of the ESI parameters would lead to activation prior to the first quadrupole, i.e. in-source activation. In the utilized instrument and given the small size of the studied system, this type of activation only marginally differs from the trap activation shown here in Figures 4, S5, S6 and S7. However, here trap activation experiments after quadrupole isolation of the precursor were performed, which provides much more controlled and comparable conditions. Following migration after in-source activation on the other hand, would lead to ambiguous data that cannot be interpreted.

The trap activation method makes sense since this minimizes spread in internal energy of ions.

In order to clarify that there is no migration prior to CID, we have added Figure S1 which shows the intact HS2SNAc (the precursor) peak in comparison to a HS6SNAc (the migration product) standard; identical source parameters have been used for this measurement and the following ones, hence in this case, no migration prior CID activation is occurring. To reflect this comment, we have adjusted the sentence on page 5 from:

“As a result, the sulfate group must change its position from the uronic acid to the adjacent GlcNAc prior to the fragmentation of the molecule.”

to

“As a result, the sulfate group must change its position from the uronic acid to the adjacent GlcNAc after the in-source ionization (Figure S1) but prior to the fragmentation of the molecule.”

This is an acceptable response.

2. To further investigate the mechanism, the authors may consider protecting acidic sites to ensure there is no mobile proton.

We thank the reviewer for this comment. The necessity of the mobile proton was discussed extensively in the previous work by Kenny et al. (reference 20 in the manuscript, Kenny, D. T.; Issa, S. M. A.; Karlsson, N. G., Sulfate migration in oligosaccharides induced by negative ion mode ion trap collision-induced dissociation. *Rapid Communications in Mass Spectrometry* 2011, 25 (18), 2611-2618.) A repetition of these experiments would not lead to further insights.

Overall, this is a good study, but it may not be novel enough to be published in *Nature Communications*.

This is a good suggestion by Reviewer 1 since it can help rule out other mechanisms such as the vehicle mechanism which relies on diffusion of the sulfate group. However, since the mechanism is stated as concerted by the author then it may be more Grotthus-like.

Reviewer #2 (Remarks to the Author):

Glycan sulfation has important biological functions such as protein binding specificity, structural integrity, and cell signaling. It may therefore be crucial and helpful to determine the native atomic site of sulfation to potentially advance biomarker discovery, diagnostics, or therapeutic design. Elucidation of a glycosaminoglycan sulfation site(s), for example, can be achieved through fragmentation-based sequencing using collision induced dissociation (CID) and traditional mass spectrometry. However, sulfate migration has been observed to occur during the CID process, which

makes structural analysis very challenging since traditional MS cannot properly separate and detect isobars. Indeed, structural rearrangement, as noted by the authors, is a well-known phenomenon in MS studies and has been studied extensively by other labs for a range of chemical systems. Hence, the overall novelty of the work is limited. In this work, Polewski et al. aim to use ion mobility mass spectrometry (IM-MS) and molecular modeling to elucidate both the isomeric products (i.e., isobars) and the inherent mechanisms of sulfate group rearrangement. Although this work explores an important and interesting phenomenon in the mass spectrometry domain, the conclusion relies on limited data. Additionally, the computational methods need further clarification.

A large number of papers have appeared on CCS modeling and the use of ML tools to compute CCS values. However, none of this is noted – even as a justification for their computational protocol, which is not novel in light of earlier work.

We agree with the reviewer that the ML tools become useful for the prediction of CCS values. Nonetheless, the training and parametrization of such models requires rigorous validations, particularly for structurally and conformationally complex molecules such as GAGs, and such work is outside of this study. Furthermore, the trajectory method used in this study, which has been implemented in MobCal 2.0 in 2023 (Haack, A. et al.; *Analyst* 2023, 148, 3257–3273) has is still under ctive development, offers a well-tested and cost-effective solution to determination of the CCSs of small molecules (recently: Gorbachev et al, *JACS*, 2025, 147, 4308). Finally, the computational cost of determining atomic charges for the potentials used in trajectory methods is marginal when compared to conformational sampling at the DFT level of theory, which reduces the need for ML-based approaches.

The concern here that remains unaddressed is that a lot of CCS modeling works have been done but not enough are noted by the authors. It is understandable that there may not have been a justification for using the software/computational tools and that the method is an arbitrary choice. However, it should be noted that different results may be produced from different sampling method and/or different DFT level of theory. This is something the authors should emphasize in the MS since the test systems are very few.

We are fully aware of various ML-based and other methods that have been developed recently to compute CCSs, which are particularly useful for calculations of large biopolymer aggregates for which the TM method become computationally expensive. However, this paper is neither a review nor a benchmark of methods to compute the CCSs, instead it applies the well-established TM algorithm to estimate CCSs values and elucidate trends in changes of CCSs for a series of molecules that overlap in size and properties with molecules TM has been developed for. When we discuss the DFT, we do not review the field of all available functionals that might or might not have relevance to this system but rather provide a context why a specific protocol has been chosen (references 17, and 32-36). Similarly, MobCal2.0 is specifically used because of physics-based TM approach to compute the CCS of small molecules in N₂ and He gases, and that the developers validated the methods for a set of relevant 36 calibrant molecules in He (Appendix D of the manual) with a RMSD of 2%, all of which is explained in reference 37. Because of this, we believe that the manuscript properly acknowledges relevant methods, and we would like to avoid citing methods that are beyond the scope of this manuscript.

Overall, rearrangement of molecular ions is a well-known problem, the experimental methods used are routine and the modeling uses approaches that have been well described previously. Given all this the paper is not well suited for communications oriented journal like Nature Communications and would be better suited for JASMS or a related MS focused journal.

We respectfully disagree with the reviewer on this point. Migration reactions have indeed been reported for phosphate groups in peptides and fucose residues in glycans. However, as mentioned above there is only a single report that describes the rearrangement of sulfates. This report is cited and discussed in the manuscript.

The sulfation pattern of glycosaminoglycans (GAGs) – often referred to as the “sulfation code” – is a defining feature of their biological function. Deciphering this code is inherently challenging and relies almost entirely on mass

spectrometry and tandem MS-based techniques. As such, the findings we present here have important implications for the structural and functional analysis of GAGs. Given the increasing interest in this area, further accelerated by developments in infection biology during the COVID-19 pandemic, we believe these insights are timely and highly relevant to the broader glycobiology community.

As I stated above, rearrangement in a general sense is a commonly observed and discussed problem for molecular ions in mass spectrometry.

Beyond these issues the following concerns should be addressed.

Comments:

1. The experimental CCS values for GlcNAc-6S and GlcNAc-3s were obtained from the arrival time distributions (ATD) of their respective ions. Typically, the ATD is influenced by the ions' conformational heterogeneity. With that said, why was the calculated (computationally produced) CCS value per system not Boltzmann-weighted with respect to the ensemble relative energies to simulate a CCS that accounts for conformation heterogeneity.

We fully agree with the reviewers that, assuming that the relevant parts of the molecular phase space is ergodic, the ATDs represent conformational average of the respective ions. However, the accurate determination of such Boltzmann-weighted CCSs would require either averaging over full conformational space or accurate calculations of the free energy of most important conformational states. Histograms of multiple 1 ns MD simulations (GNF2-xTB level of theory) are presented in the SI, but neither such trajectories represent a fully converged sampling, nor the level of theory offers accurate representation of the PES. On the other hand, averaging over most stable minima on the PES derived at higher level of theory (DFT) suffers from the shortcomings of the harmonic approximation: it underestimates the population of extended conformers. Thus, while we report the absolute values of the most stable (and relevant conformers), our analysis focuses more on changes to the CCSs between different isomers and labels.

I believe that authors can just Boltzmann-weighted average over DFT electronic energies obtained for the ensemble of conformations optimized. These are the same conformations used to calculate the CCS values, if I am not mistaken. From my understanding, an ATD curve shape is influenced by different conformers or isomers populated. I am curious to see what the weighted average CCS is when conformers and isomers are all taken into account.

First, we also want to note we do not aim at predicting CCSs of unknown ions, but rather explaining why one of the two isomers adopts a more extended conformation in the gas phase. In broader scope, the estimated Boltzmann-weighted CCSs depend on the quality of the underlying DFT free energies, herein calculated using harmonic approximation at 300K, and the quality of sampling techniques. With respect to these points, we already noted that CREST was biased towards more compact conformations of GlcNAc-3S-ProA, and the harmonic approximation is known to underestimate the stability of the extended structure due to incorrect assessment contributions of low-energy vibrational modes to the free energy. Finally, the ions do not exist as fixed geometries but they experience thermal motions around the minimum, which also have intrinsic distribution of CCSs. In effect, all these errors might propagate to Boltzmann-averaged CCS which can cause much larger errors than those derived from a single most stable minima.

That being said, we estimated the weight-averaged CCSs for the ions discussed in this work, and they indeed depend on the dataset being used. In details, for GlcNAc-6S-ProA the Boltzmann-averaged CCSs (based on data in Figure 2C) are equal to 152.3, 153.3, and 153.5 Å² for, respectively, from MD, CREST, and combined datasets, while for GlcNAc-3S-ProA the respective averages are 170.9, 152.4, and 164.8 Å². The experimental ^{DT}CCS_{He} are equal to 151 and 168 Å² for GlcNAc-6-ProA, and GlcNAc-3S-ProA, respectively, which are relatively close to the Boltzmann-weighted CCSs of MD-derived conformers, and combined datasets. In the manuscript, however, we used CCSs of the lowest free energy conformers, which have CCSs of 152 and 173 Å², and they are also relatively close to the measured

$^{DT}CCS_{He}$ s. Thus, the assignment of the experimental CCSs to more compact GlcNAc-6-ProA, and more extended GlcNAc-3S-ProA remains unchanged.

Finally, The two sampling techniques used for structure assignments favor (indicated as a bias in the manuscript) either GlcNAc-6S or GlcNAc-3s (Figure 2C). Were two sampling techniques implemented from the start of this work or was it added after to improve the agreement with experiment? Since the purpose of using two different sampling techniques is to sample a broader conformational space, why not use a conformation generator with no energetic preference. This approach would remove sampling bias.

In principle, conformational sampling without use of energy function as a bias means constitutes a systematic or a random search. However, these two classical approaches are well known to fail when a large number of degrees of freedom becomes a factor. In the context of this work, we decided to combine two conformational techniques to explore larger conformational space which is not uncommon in complex molecular systems. For example of such approach, please see Scutelnic et al, J. Am. Chem. Soc. 2018, 140, 24, 7554–7560.

The success in capture IM-MS relevant ions is improved with an increase in the number of conformations sampled

The response seems to be incomplete. Nevertheless, we do not agree with this statement, the point of conformational sampling methods is to reduce the number of conformations that needs to be sampled to obtain an answer.

2. To help with reproducibility, please indicate which charge scheme was used in Gaussian16 to obtain atomic partial charges for the CCS calculation using the trajectory method.

The protocol to calculate the charges is clearly explained in the original publication (Haack, A. et al; Analyst 2023, 148, 3257–3273). We have clarified in the methods and SI sections that we have used the same partition scheme and level of theory to compute the atomic charges:

“To perform CCS calculations using MobCal 2.0, we computed atomic charges using CHELPG partition scheme available in Gaussian16 using recommended ω B97X-D3/def2-TZVPP level of theory.”

This is an adequate response.

3. Line 322 in the manuscript states that TW-IMS (Traveling Wave Ion Mobility Spectrometry?) settings were used, however, throughout the text the DT superscript is attached to CCS (i.e., $^{DT}CCS_{He}$). If not mistaken, DT is for drift tube ion mobility spectrometry. Thus, were the experimental CCS values obtained from TWIMS or DTIMS?

The sub- and superscripts are descriptors of the measurements conditions and follow established community recommendations. For details see doi: 10.1002/mas.21585.

Measurements were performed on both, a standard TW Synapt G2-S and a modified DT Synapt G2S. The presented ATDs are from TW-IMS measurements in N₂ as these are usually better resolved; the reported $^{DT}CCS_{He}$ values were derived from direct CCS measurements on a DT instrument using the stepped field method. Helium was chosen as a drift gas as it can be described more accurately using theoretical calculations. For details see doi: 10.1002/mas.21585.

So the DTIMS data were used as calibration standards for assigning CCS values to the TWIMS ATD? If this is true then the authors should make this more clear for those readers who are not knowledgeable in mass spectrometry procedures/technologies.

This is not correct; the shown ATDs were measured on a TW-IMS instrument due to the higher resolution, but $^{DT}CCS_{He}$ values were extracted through a separate measurement on a DT-IMS instrument, which gives more accurate, absolute CCS values through the stepped field method described in source 41 and the methods section.

The utilized nomenclature may not have been described properly in the previous version. To address this, we included a publication with community guidelines on the reporting of CCSs as the reference 43. (Gabelica, et al. (2019), Recommendations for reporting ion mobility Mass Spectrometry measurements. Mass Spec Rev, 38: 291-320). To clarify it, we have added the following sentence at the end of the MS and IM-MS measurements section in the Methods:

“The notation follows the community guidelines on the reporting of CCSs.⁴³”

4. Are the conformation sampling temperatures of 400K and 450K similar to the temperatures that were experimentally observe during the CID event or drift region? Or were these temperatures selected arbitrarily to enhance s conformational sampling?

In the present case, measurements were performed at ambient conditions using room temperature drift gas. However, the effective temperature of the ions is certainly higher, especially after CID. Measuring this effective temperature is non-trival and beyond the scope of this work. In order to still sample a reasonable range of conditions, the temperatures were arbitrarily chosen.

This is a reasonable response.